# Impact of Zirconia and Titanium Implant Surfaces of Different Roughness on Oral Epithelial Cells

**DOI:** 10.3390/dj14010030

**Published:** 2026-01-04

**Authors:** Marco Aoqi Rausch, Zhiwei Tian, Vera Maierhofer, Christian Behm, Christian Ulm, Erwin Jonke, Raphael S. Wagner, Benjamin E. Pippenger, Bin Shi, Xiaohui Rausch-Fan, Oleh Andrukhov

**Affiliations:** 1Competence Center for Periodontal Research, University Clinic of Dentistry, Medical University of Vienna, 1090 Vienna, Austria; marco.rausch@meduniwien.ac.at (M.A.R.); 840574999tzw@gmail.com (Z.T.); vera.maierhofer@meduniwien.ac.at (V.M.); christian.behm@meduniwien.ac.at (C.B.); 2Clinical Division of Oral Surgery, University Clinic of Dentistry, Medical University of Vienna, 1090 Vienna, Austria; christian.ulm@meduniwien.ac.at; 3Clinical Division of Orthodontics, University Clinic of Dentistry, Medical University of Vienna, 1090 Vienna, Austria; erwin.jonke@meduniwien.ac.at; 4Institut Straumann AG, 4052 Basel, Switzerland; raphael.wagner@straumann.com; 5Department of Surgical Research, AnaPath, 4410 Liestal, Switzerland; bpippenger@anapath.ch; 6Department of Oral and Maxillofacial Surgery, The First Affiliated Hospital of Fujian Medical University, Fuzhou 350005, China; drshibin@163.com; 7Center for Clinical Research, University Clinic of Dentistry, Medical University of Vienna, 1090 Vienna, Austria; xiaohui.rausch-fan@meduniwien.ac.at; 8Division of Periodontology, University Clinic of Dentistry, Medical University of Vienna, 1090 Vienna, Austria

**Keywords:** dental implants, titanium, zirconia, soft tissue, oral epithelial cells, surface roughness, interleukin-8, integrins, epithelial attachment

## Abstract

**Background/Objectives**: Formation of tight contacts between oral soft tissue and dental implants is a significant challenge in contemporary implantology. An essential role in this process is played by oral epithelial cells. In the present study, we investigated how titanium and zirconia surfaces with different roughness influence various parameters of oral epithelial cells in vitro. **Methods**: We used the human oral squamous carcinoma Ca9-22 cell line and cultured them on the following surfaces: machined smooth titanium (TiM) and zirconia (ZrM) surfaces, as well as sandblasted and acid-etched titanium moderately rough (SLA) and zirconia (ZLA) surfaces. Cell proliferation/viability was measured by CCK-8 assay, and cell morphology was analyzed by fluorescent microscopy. The gene expression of interleukin (IL)-8, intercellular adhesion molecule (ICAM)-1, E-cadherin, integrin (ITG)-α6, and ITG-β4 was measured by qPCR, and the content of IL-8 in conditioned media by ELISA. **Results**: At the initial culture phase, cell proliferation was promoted by rougher surfaces. Differences in cell attachment were observed between machined and moderately rough surfaces. Machined surfaces were associated with slightly higher IL-8 levels (*p* < 0.05). Furthermore, both ZLA and SLA surfaces promoted the expression of (ITG)-α, ITG-β4, and ICAM-1 in Ca9-22 cells (*p* < 0.05). Surface material had no impact on the investigated parameters. **Conclusions**: Under the limitations of this in vitro study, some properties of oral epithelial cells, particularly the immunological and barrier function, are moderately modified by roughness but not by material. Hence, the roughness of the implant surface might play a role in the quality of the peri-implant epithelium.

## 1. Introduction

The establishment and maintenance of healthy soft tissue around dental implants are critical factors in ensuring the long-term success of implant-supported restorations. A stable peri-implant soft tissue seal serves as a biological barrier, protecting underlying bone from microbial infiltration and subsequent inflammatory destruction [1,2,3,4]. The junctional epithelium plays a crucial role in this defense mechanism, forming an attachment to the implant surface through specialized adhesion molecules [5]. Failure to achieve adequate epithelial attachment and soft tissue integration can lead to increased susceptibility to peri-implant diseases, including peri-implant mucositis and peri-implantitis [2].

Implant surface characteristics, such as topography, roughness, hydrophilicity, and coating, are considered as key determinants of the biological processes at the implant-host interface [6,7]. Particularly, modification of titanium (Ti) implants resulting in moderately rough hydrophilic implant surfaces provides a superior outcome in terms of osteointegration [8]. At the cellular level, surfaces with these characteristics enhance the response of osteoblasts, the primary cell type responsible for bone formation [9,10]. In contrast, the optimal surface treatment for the integration of mucosal tissue is still to be found and is a topic of current research [11]. The soft tissue seal around the implant should be as effective as that in natural teeth in terms of protecting underlying bone from microorganisms and environmental factors, but this is difficult to achieve [12]. Particularly, junctional epithelium, the main component of antimicrobial defence at the dentogingival junction, is markedly different between implants and natural teeth [13]. A recent study has suggested that surfaces with varying roughness promote the response of specific soft tissue cells, and this strategy may be used to improve the soft tissue barrier [14]. Implant material could be another factor influencing implant contact with soft tissue. A recent systematic review and meta-analysis suggest that zirconia (Zr) shows a superior outcome in some aspects of soft tissue response compared to Ti [15].

Numerous studies addressed the impact of implant surface modifications on osseointegration [16,17]. Compared to that, the question of how material type and surface roughness influence soft tissue behavior, particularly epithelial cell responses, has been less investigated [1,18,19]. A recent study showed that a smooth titanium surface with an arithmetical mean height (Sa) value of 0.11 µm is superior compared to rougher surfaces with Sa of 0.39, 1.33, and 3.34 µm regarding epithelial cell viability and cytotoxicity [20]. In contrast, another study showed that a titanium surface with a Sa of about 0.4 µm inhibits the proliferation of oral epithelial cells after a prolonged time compared to a machined surface [14]. In our former study, only minimal differences between Ti surfaces with various roughness and hydrophilicity were found in terms of epithelial cell viability and gene expression. Another study showed an improved adhesion and higher expression of the integrin-β4 (ITG-β4) gene in epithelial cells grown on the machined compared to the rough surface [21]. A recent study comparing Ti and Zr surfaces revealed that epithelial cells exhibit superior adhesion and ITG-β4 expression on the Ti surface, as well as increased migration on the Zr surface [22]. Another study on immortalized epithelial OKF6-TERT2 cells showed that a rougher Zr surface leads to higher expression of ITG-α6 and ITG-β4 and superior hemidesmosome formation than a smoother Ti surface [23]. Nevertheless, the impact of surface material and characteristics on epithelial cells still needs to be investigated. Our former study showed that application of sand-blasting and acid etching to titanium and zirconia surfaces results in different surface topography, and particularly roughness [24]. Various topographies, produced by combinations of distinct materials and surface treatment, might produce different effects on epithelial cells and thus soft tissue healing. Systematic studies on the effects of various surface parameters on the behaviour of oral epithelial cells on the implant surface are highly desirable.

The present study was therefore designed to investigate the effects of implant material and surface roughness on the behaviour of oral epithelial cells in vitro. Specifically, we compared the response of epithelial cells to smooth zirconia machined surface (ZrM), moderately rough zirconia sand-blasted acid-etched surface (ZLA), smooth titanium machined surface (TiM), and moderately rough titanium sandblasted acid-etched surface (SLA). We used in our study the oral squamous carcinoma Ca9-22 cell line, which resembles the majority of the properties of oral epithelium and does not exhibit early senescence, which is typical for primary cells [25]. The impact of various surface roughness and material on cell proliferation, morphology, and the expression of key markers associated with adhesion, immunological function, and epithelial barrier formation was assessed.

## 2. Materials and Methods

### 2.1. Preparation of Experimental Implant Discs

Four types of experimental implant discs were manufactured by Institut Straumann AG (Basel, Switzerland): smooth Ti machined surface (TiM); smooth ZrO_2_ machined surface (ZrM); moderately rough Ti sandblasted large-grit acid-etched (SLA) surface; moderately rough ZrO_2_ surface (ZLA). Details on samples’ preparation and their characteristics in terms of roughness and wettability are described in our previous study [24]. Briefly, commercial-grade 4 pure titanium was used for titanium surfaces, and Yttria-stabilized zirconia (3Y-TZP) was used for zirconia surfaces. Both machined surfaces, TiM and ZrM, were fabricated through a machining and grinding process. Moderately rough SLA and ZLA surfaces were produced by sand blasting with corundum and acid etching. The implant discs were 15 mm × 1 mm in size and fit into a well of a 24-well tissue culture plate. The following Sa values were measured previously for these surfaces [24]: ZrM, 0.12 µm; ZLA, 0.45 µm; TiM, 0.07 µm; SLA, 1.12 µm. The contact angle, which is a static contact angle measured with a 1 µL drop of ultrapure water, was 72.3° for ZrM, 119.5° for ZLA, 80.5° for TiM, and 107.0° for SLA [24]. In the present study, surfaces from the same batch were used for the experiments.

### 2.2. Cell Culture

To examine the effects of different implant surfaces on epithelial cells, we used commercially available oral squamous cell carcinoma cells (Ca9-22) derived from human gingival carcinoma. According to cell bank information (Japanese Collection of Research Bioresources Cell Bank, JCRB0625), the lifespan, function, and morphology of Ca9-22 cells are indicated as epithelial-like. Despite some limitations, this cell line is often used as a model of oral epithelium [26,27]. Modified Eagle’s minimum essential medium (MEM, Gibco^®^, Carlsbad, CA, USA) supplemented with 10% fetal bovine serum (FBS, Gibco, Carlsbad, CA, USA), penicillin (100 U/mL) and streptomycin (100 μg/mL) was used for the culture of Ca9-22 cells. The culturing was performed in a humidified atmosphere containing 5% CO_2_, at 37 °C [28]. In the experiments, cells from passages 4 to 7 were used.

### 2.3. Cell Growing on the Surfaces of Implant Discs

Implant discs were placed in a 24-well cell culture plate. Ca9-22 cells at a concentration of 2 × 10^4^/mL were seeded on different implant surfaces in 1 mL of conditional medium as described above. Cells seeded on tissue culture plastic at a similar density served as a control. Cells were grown on the implant surfaces for up to 7 days. On day 4, the culture medium was replaced with 500 µL of fresh, fully supplemented MEM.

### 2.4. Cell Proliferation/Viability

Cell proliferation and viability were determined using the Cell Counting Kit-8 (CCK-8, Dojindo Molecular Technologies Inc., Gaithersburg, MD, USA) as previously described [28]. After 2 and 7 days, 100 µL of CCK-8 reagent solution was pipetted into each well, and the cells were incubated at 37 °C for two hours. Afterwards, 100 µL of the solution was transferred into a new 96-well plate, and the optical density was measured using the Synergy HTX Multi-Detection Reader (Synergy HTX, BioTek Instruments, Winooski, VT, USA) at an absorbance of 450 nm. CCK-8 experiments were performed at least in triplicate. Cells grown on tissue culture plastic served as a control.

### 2.5. Focal Adhesion Staining and Cell Morphology

Analysis of cell morphology and attachment was performed after 1, 3, and 7 days of culture using the Focal Adhesion Staining Kit (FAK100, Millipore, Burlington, MA, USA) as previously described [29,30]. Cells were fixed with 4% paraformaldehyde solution (Nordic Biosite, RBB-A10YNP-500, 15 min at room temperature) and permeabilized with 0.1% Triton X-100 (5 min at room temperature). Ca9-22 cells were then incubated in a blocking solution containing 1% bovine serum albumin for 30 min, followed by an incubation with primary Anti-Vinculin antibody (1:200, Millipore, Burlington, MA, USA) for 60 min. After washing steps, secondary FITC-conjugated goat anti-mouse IgG Antibody (1:200, AP124F, Sigma-Aldrich, St. Louis, MO, USA) and TRITC-conjugated phalloidin (1:200, Millipore, Burlington, MA, USA), were incubated simultaneously for 60 min. Nucleus counterstain was performed with 4′, 6-diamidino-2-phenylindole (DAPI, 1:800, Sigma-Aldrich, St. Louis, MO, USA) at room temperature for 5 min. Cell morphology and adhesion were observed using the ECHO Revolve fluorescence microscope (Echo, San Diego, CA, USA), and images were captured with the appropriate fluorescence filters using a 10× objective magnification.

### 2.6. Quantitative Real-Time PCR

Resulting gene expression levels of different functional proteins in Ca9-22 cells were measured by quantitative polymerase chain reaction (qPCR). Cell lysis was prepared at 7 days of cell incubation, and extraction of mRNA, reverse transcription into cDNA and qPCR were performed using the TaqMan Gene Expression Cells-to-CT kit (Applied Biosystems, Foster City, CA, USA) according to the manufacturer’s protocol [31]. The thermocycler Biometra One (Analytik Jena, Jena, Germany) was used for reverse transcription and QuantStudio3 (Applied Biosystems, Foster City, USA) for qPCR. TaqMan Gene Expression Assays (Applied Biosystems, Foster City, USA) with the following ID numbers were ordered for paired reactions: all from Applied Biosystems TaqMan: Cadherin (CADH)-1: Hs01023894_m1; Interleukin (IL)-8: Hs00174103_m1; Integrin alpha (ITGα)-6: Hs01041011_m1; Integrin beta (ITGβ)-4: Hs00173995_m1; Intercellular adhesion molecule (ICAM)-1: Hs00164932_m1 Glyceraldehyde 3-phosphate dehydrogenase (GAPDH): Hs99999905_m1 was used as a house-keeping gene). After carrying out the reactions in technical duplicates at 95 °C for 10 min, 40 cycles, each for 15 s at 95 °C and for 1 min at 60 °C, the 2^−ΔΔCt^ method was applied to quantify the gene expression levels according to the following formula: ΔΔCt = (Ct^target^ − Ct^GAPDH^)_sample_ − (Ct^target^ − Ct^GAPDH^)_control_. The n-fold expression of the target genes was determined in relation to the untreated control. Glycerinaldehyd-3-phosphat-Dehydrogenase (GAPDH) served as an endogenous control.

### 2.7. Interleukin-8 ELISA

The level of IL-8 protein in the conditioned media after 2 days of culturing was determined by a commercially available ELISA (Cat. Nr. 88-8086-88, Invitrogen, Waltham, MA, USA) as described in our recent study [32]. The absorbance measurements at the wavelength of 450 and 570 nm were done using a microplate reader (BioTek Instruments, Winooski, VT, USA) and the curve fitting and output calculation was done using the Gen5 software (Version 2.09, BioTek Instruments, Winooski, VT, USA) were used for the absorbance measurements and data analysis, respectively.

### 2.8. Statistical Analysis

Statistical differences between various groups were assessed using the Friedman test and the Wilcoxon test was used for the pairwise comparisons. The differences were considered to be statistically significant when *p*-values were less than 0.05. Data are presented as mean ± SD of five to six biological replicates. The presentation of SD rather than s.e.m. was chosen because SD reflects the dispersion of the data from the mean and is a preferred form of data presentation [33]. All statistical analysis were made with the SPSS 26.0 software (IBM, Armonk, NY, USA) and the graphs were made using the GraphPad Prism 10 (GraphPad Software, Boston, MA, USA).

## 3. Results

### 3.1. Proliferation/Viability of Ca9-22 Cells Grown on Various Surfaces

Proliferation/viability of Ca9-22 cells grown on the various surfaces for 2 and 7 days is presented in Figure 1. At day 2, proliferation/viability of Ca9-22 cells grown on moderately rough surfaces was slightly but significantly higher compared to smooth surfaces of similar material, i.e., ZrM vs. ZLA and TiM vs. SLA (*p* < 0.05). Moreover, the proliferation of cells on the SLA surface was significantly higher compared to ZLA. At day 7, cell proliferation and viability were slightly but significantly higher on ZrM compared to ZLA, and SLA compared to TiM. No significant difference was observed between the surfaces of different materials at this time point.

### 3.2. Morphology of Ca9-22 Cells Grown on Various Surfaces

The morphology of Ca9-22 cells grown on different surfaces at days 1, 3, and 7 of culture and visualized by fluorescent microscopy is presented in Figure 2. At day 1, single cells were randomly attached to all surfaces. Some of the cells grown on the ZrM and TiM surfaces exhibited a slightly prolonged shape. At day 3, cells started to form clusters; these clusters had a prolonged shape on TiM surfaces and a random shape on ZLA and SLA surfaces. The morphology of clusters observed on the ZrM surface was intermediate between these two models. At day 7, cells were near confluent on all surfaces. Still, cells grown on both machined surfaces had a slightly prolonged morphology compared to moderately rough surfaces.

### 3.3. Morphology of Ca9-22 Cells Grown on Various Surfaces

#### 3.3.1. Interleukin-8

The impact of different surfaces on the gene and protein expression of IL-8 in Ca9-22 cells grown on different surfaces is shown in Figure 3. Cells grown on moderately rough ZLA surfaces exhibited significantly lower gene expression and protein production of IL-8 compared to the smooth ZrM surface. No differences in IL-8 gene expression and protein production were observed between TiM and SLA surfaces, as well as between surfaces from different materials.

#### 3.3.2. Integrin α6 and Integrin β4

The gene expression of ITGα6 and ITGβ4 in Ca9-22 cells grown on different surfaces for 7 days is shown in Figure 4. In both materials, the gene expression of these integrin subunits was significantly higher in cells grown on moderately rough ZLA and SLA surfaces compared to smooth machined ZrM and TiM surfaces, respectively. No significant differences between similarly treated surfaces of different materials were observed.

#### 3.3.3. E-Cadherin and ICAM-1

Figure 5 shows the gene expression of E-cadherin and ICAM-1 in Ca9-22 cells grown on different surfaces. No significant differences in E-cadherin gene expression were found between different surfaces. However, a tendency for slightly lower E-cadherin gene expression on ZLA and SLA surfaces compared to ZrM and TiM surfaces, respectively, was observed. ICAM-1 gene expression was significantly higher in cells on the ZLA surface compared to the ZrM surface. A similar tendency was observed for SLA and TiM surfaces, but the differences were not statistically significant. Cells grown on similarly treated surfaces of different materials exhibited no significant differences in ICAM-1 gene expression.

## 4. Discussion

This study investigated the impact of implant material and surface roughness on oral epithelial cells in vitro. We used the oral squamous carcinoma Ca9-22 cells, which reflect many properties of oral epithelium and are often used in research despite some limitations [26,27]. We focused on the proliferation, viability, attachment, morphology, and expression of proteins involved in the inflammatory response, epithelial barrier function, and transepithelial neutrophil migration. These cells were grown on either zirconia or titanium surfaces with various treatments: machining to produce smooth surfaces and sand-blasting/acid-etching to produce moderately rough surfaces. It should be noted that, despite a similar roughening protocol being applied to Zr and Ti surfaces, the resulting surfaces exhibited strikingly different Sa values, 0.45 µm for ZLA and 1.12 µm for SLA surfaces [24]. This fact should be taken into account when interpreting the data.

The proliferation and viability of epithelial cells were assessed using the CCK-8 assay, which measures mitochondrial activity and indicates the proliferation of viable cells [34]. At day 2, when cells on all surfaces were not yet confluent and were presumably in their growing phase, proliferation and viability were significantly higher on moderately rough surfaces compared to smooth surfaces. Moreover, the proliferation/viability of Ca9-22 cells on SLA surface with a Sa value of about 1.12 µm was significantly higher than that on ZLA surface with a Sa value of 0.45 µm. This finding suggests that, at least at the initial phase, the proliferation/viability of oral epithelial cells might be stimulated by an increased surface roughness. In contrast, after 7 days of culture, when cells were almost confluent, no such clear relationship between surface roughness and proliferation/viability was observed. A stimulating effect of surface roughness on epithelial cell proliferation has already been reported in a study on the HaCat cell line comparing the anodized Ti surface with the machined Ti surface [35]. However, some previous studies also reported an opposite effect [20,36]. One study showed a higher proliferation of human gingival keratinocytes growing on a smooth titanium surface (Sa 0.11 µm) compared to minimally rough (Sa 0.39 µm), moderately rough (Sa 1.33 µm), and rough (Sa 3.34 µm) surfaces [20]. Another study showed an improved proliferation of human primary gingival keratinocytes and epithelial progenitor cells on Ti machined surface with the arithmetic average roughness (Ra) of 0.3–0.6 µm compared to other Ti surfaces with Ra of 1.2 and 2.0 µm [36]. The reasons for these differences may include variations in surface treatment methods, the cell line used, and experimental protocols. The implant surface material appears to have no noticeable effect on epithelial cell proliferation. We did not find any differences between machined surfaces from zirconia and titanium. We observed higher proliferation and viability for the titanium SLA surface than for the zirconium ZLA surface after 2 days (Figure 1). However, this difference could be attributed to the higher roughness of the SLA surface [24]. No effect of surface material on epithelial cell proliferation was also reported by a study comparing zirconia and titanium surfaces with the Ra of about 0.15 µm [37].

We also observed significant differences in epithelial cell attachment and morphology on surfaces treated with different methods. Cells grown on smooth surfaces displayed an elongated morphology and tended to align along a single axis, while cells on moderately rough surfaces exhibited a more random orientation and broader spreading across the surface. A directed orientation of epithelial cells may be attributed to the presence of microscopic grooves on ZrM and TiM surfaces, which are presumably formed during the machining process and can be observed in the SEM images from our earlier study [24]. A similar phenomenon was observed in previous studies on gingival fibroblasts [29,38]. A recent study also observed the effect of surface roughness on the attachment and spreading of different types of cells, including the human gingival epithelioid cell line [39]. Thus, the treatment of the implant surface may alter cell attachment, which in turn could impact the quality of the soft tissue seal around the implant.

We further focused on the gene expression and protein production of IL-8 by epithelial cells grown on different surfaces. IL-8 is a potent chemokine involved in the recruitment of leukocytes [40]. The level of IL-8 in peri-implant crevicular fluid is substantially increased at sites with peri-implantitis [41]. We found that cells grown on a moderately rough ZLA surface produced significantly less IL-8 compared to the ZrM surface. For titanium surfaces, such a relationship was not observed, although gene expression was also tendentially lower on moderately rough SLA than on machined TiM surface. In our previous study on gingival fibroblasts, we found that under certain conditions, cells growing on moderately rough surfaces produce fewer inflammatory mediators compared to those on machined surfaces [24]. Therefore, a moderately rough surface might be associated with a generally lower inflammatory response of soft tissue cells.

Integrin subunits α6 and β4 are essential elements of hemidesmosomes and are involved in the attachment of oral epithelium to the basement membrane [42,43]. Recent studies showed that these integrin subunits are also involved in the attachment of epithelium to the implant surface [2,44]. We observed that the gene expression of these two integrin subunits was promoted by moderately rough surfaces (Figure 4). This fact implies that moderately rough surfaces might provide a better attachment of the oral epithelium. This assumption is supported by a recent study on dogs, which showed that roughened-surface implants resulted in higher supracrestal soft tissue height than machined implants [45]. Another study compared the expression of ITG-α6 and ITG-β4 proteins and hemidesmosome formation and found that a rough Zr surface had a superior effect on these parameters than a smooth Ti surface [23]. Interestingly, the creation of nanocrystals on the smooth surface can also upregulate ITG-β4 expression in oral epithelial cells [46]. However, these findings contradict a previous report using rat oral epithelial cells, which showed decreased ITG-β4 protein expression on a rougher Ti surface [21]. The differences among studies could presumably be due to the use of different cell lines and the experimental design. We did not observe any effect of implant material on the gene expression of integrin subunits. This finding is consistent with a recent study on rats, which showed a similar localization of ITG-β4 around Ti and Zr implants [44].

E-cadherin is a key adhesion protein involved in maintaining barrier function [47]. Degradation of E-cadherin is considered as an essential bacterial strategy for disrupting the epithelial barrier [48,49]. ICAM-1 is a key protein in regulating trans-epithelial migration of leukocytes [50]. We found only a minimal effect of implant surface roughness on the gene expression of these two proteins in oral epithelial cells. Particularly, moderately rough surfaces seemed to slightly decrease the expression of E-cadherin, although without statistical significance. ICAM-1 gene expression was increased on moderately rough ZLA compared to ZrM surface; a similar tendency was observed between SLA and TiM. It should be noted that these differences in gene expression of E-cadherin and ICAM-1 were rather small, and their physiological relevance should still be clarified.

Various implant surface modification strategies were considered to improve soft tissue seal [4,51]. This is quite a challenging task because the implant surface should promote soft tissue attachment on the one hand and minimize bacterial colonization on the other hand [51,52]. The data obtained in the present study showed a complex relationship between surface material and its treatment on epithelial cells. When interpreting our data, it should be noted that the ZLA surface exhibits significantly lower roughness than the SLA surface despite a similar surface treatment procedure. Machined surfaces promote cell attachment in a particular direction, which might promote the formation of isotropic tissue. However, it is not known if such tissues will provide a better sealing contact between the epithelium and the implant. Both moderately rough surfaces, irrespective of the material, improve the characteristics of epithelial cells responsible for their attachment to the implant surface. However, slightly better characteristics regarding inflammatory response were observed only for the ZLA surface.

Although a moderately rough surface seems to elicit a superior response of oral epithelial cells, it is questionable whether it would lead to a more stable soft tissue seal in the long term. It should be considered that increased roughness facilitates bacterial adhesion and colonization [53,54], which may negatively affect soft tissue stability. The use of more sophisticated 3D models could be an important tool for investigating further soft tissue formation around various implant materials [55,56,57,58]. Furthermore, different implant surface treatment protocols might be applied to specifically improve the integration of the epithelial junction and soft connective tissue [14].

Certain limitations of this study should be acknowledged. Firstly, the in vitro model used Ca9-22 cells, a squamous carcinoma cell line, which are suitable for controlled analysis but may not fully reflect the properties of the native oral epithelium. Most of the effects found in this study are moderate, and the results regarding gene expression of adhesion proteins and integrin subunits should still be verified at the protein level. Further, the study does not consider the complex environment around the implant, particularly the presence of microbiome, different cell types, and the inflammatory environment. Some bacterial components, particularly *Porphyromonas gingivalis* gingipains, may impair epithelial cell attachment to the Ti implant surface [59]. To address these limitations, future research should involve more sophisticated models that better replicate in vivo conditions.

## 5. Conclusions

The present study demonstrated that surface roughness exerts a more pronounced influence than implant material on the behaviour of oral epithelial cells. While both titanium and zirconia surfaces supported comparable levels of cell viability, moderately rough surfaces were associated with enhanced early proliferation, distinct morphological adaptations, lower inflammatory response, and upregulation of integrin subunits. These molecular and morphological responses suggest that moderately rough surfaces promote stronger epithelial attachment and may contribute to the development of a stable soft tissue seal. Despite these insights, it must be recognised that this study was limited to an in vitro model using Ca9-22 cells and did not account for the complex interactions present in the oral environment. Therefore, further investigations using primary epithelial cells and in vivo models are warranted to verify these observations and to determine how surface modifications can best be optimised to enhance peri-implant soft tissue integration.

## Figures and Tables

**Figure 1 dentistry-14-00030-f001:**
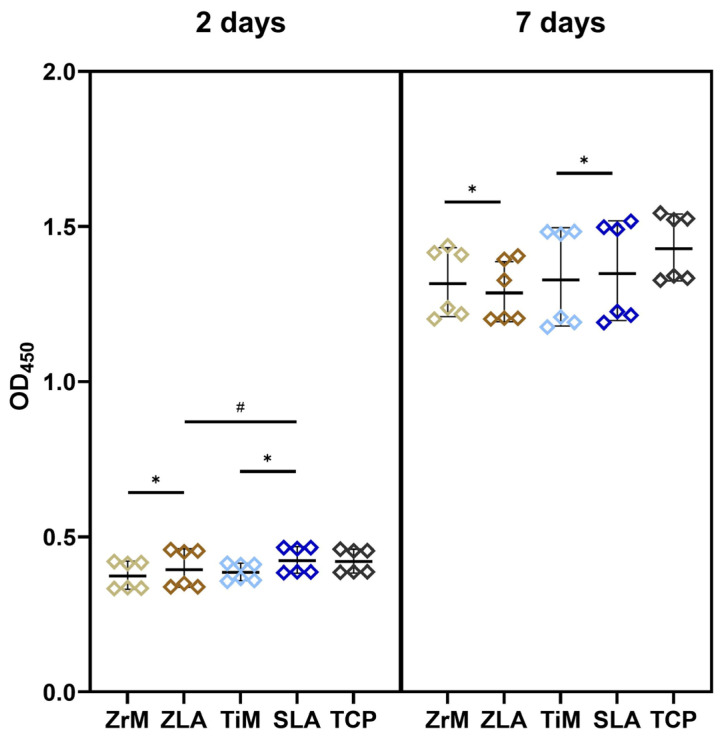
Proliferation/viability of oral epithelial cells grown on different surfaces. Ca9-22 cells were cultured in one of the following surfaces: smooth machined zirconia (ZrM), sand-blasted acid-etched moderately rough zirconia (ZLA), smooth machined titanium (TiM), sand-blasted acid-etched moderately rough titanium (SLA) surfaces. Cells cultured on the tissue culture plastic (TCP) served as a reference. The cell-counting kit-8 (CCK-8) method was used to measure cell proliferation/viability after 2 (**left panel**) and 7 (**right panel**) days of culture. The *y*-axis shows the optical density values measured at 450 nm (OD_450_). Each point represents a value obtained in an individual experiment; lines and error bars show mean and SD, respectively. *—significant difference between ZrM vs. ZLA and TiM vs. SLA, *p* < 0.05. #—significantly different between ZLA vs. SLA.

**Figure 2 dentistry-14-00030-f002:**
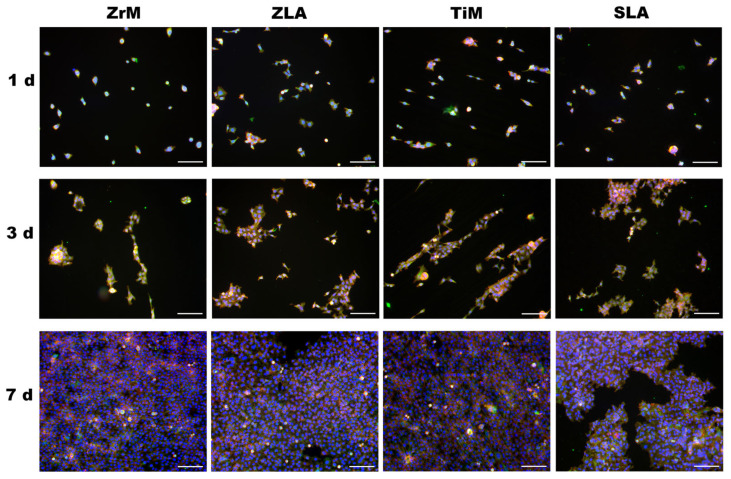
Fluorescence microscopy images of oral epithelial cells cultured on a smooth machined ZrO_2_ surface (ZrM), a moderately rough ZrO_2_ surface (ZLA), a smooth machined Ti surface (TiM), and a moderately rough, sand-blasted, large-grain acid-etched Ti surface (SLA). Ca9-22 cells were cultured for 1, 3, or 7 days; TRITC-conjugated phalloidin (red), anti-vinculin antibody in combination with the secondary fluorescein isothiocyanate (FITC, green)-conjugated antibody, and 40,6-Diamidin-2-phenylindol (DAPI, blue) were used to visualize F-actin, focal adhesions, and the nuclei, respectively. Images show one representative experiment and were taken using a 10× objective. Scale bars correspond to 100 µm.

**Figure 3 dentistry-14-00030-f003:**
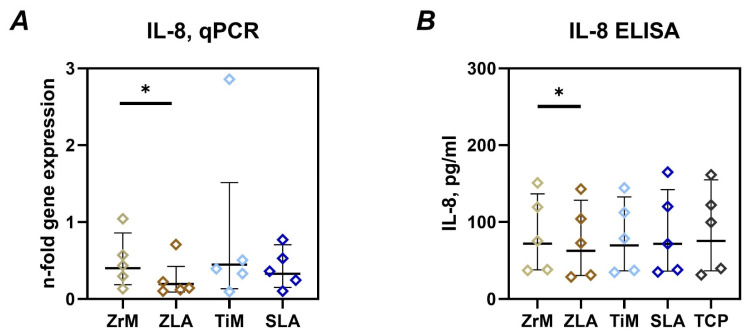
Gene expression of IL-8 and release of IL-8 protein by oral epithelial cells cultured on different surfaces. Ca9-22 cells were cultured on smooth machined zirconia (ZrM), sand-blasted acid-etched moderately rough zirconia (ZLA), smooth machined titanium (TiM), and sand-blasted acid-etched moderately rough titanium (SLA) surfaces for 7 days. (**A**)—gene expression of IL-8 measured by qPCR and presented as the n-fold gene expression compared to tissue culture plastic (TCP). This parameter was calculated using the 2^−ΔΔCt^ method, using GAPDH as the housekeeping gene. (**B**)—concentration of IL-8 in the conditioned media after 7 days of culture measured by ELISA. Data are presented as mean and standard deviation (SD); each point represents a value obtained in an individual experiment. *—significant difference between ZrM vs. ZLA, *p* < 0.05.

**Figure 4 dentistry-14-00030-f004:**
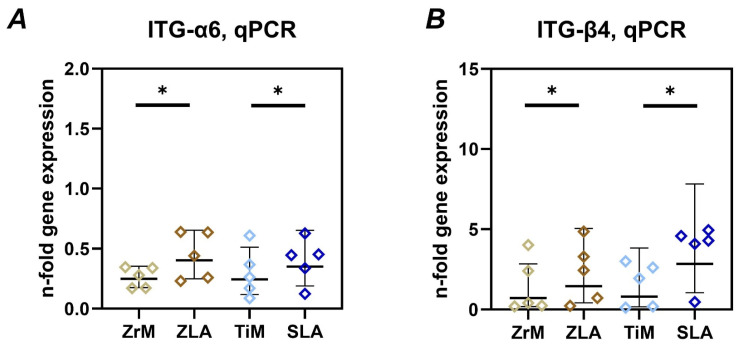
Gene expression of integrin α6 (ITG-α6) and integrin β4 (ITG-β4) subunits in oral epithelial cells cultured on different surfaces. Ca9-22 cells were cultured on smooth machined zirconia (ZrM), sand-blasted acid-etched moderately rough zirconia (ZLA), smooth machined titanium (TiM), and sand-blasted acid-etched moderately rough titanium (SLA) surfaces for 7 days. The gene expression of integrin subunits ITG-α6 (**A**) and ITG-β4 (**B**) was determined by qPCR. The data are presented as n-fold expression relative to tissue culture plastic (n = 1), calculated by the 2^−ΔΔCt^ method and using GAPDH as the housekeeping gene. Data are presented as mean and standard deviation (SD); each point represents a value obtained in an individual experiment. *—significant difference between ZrM vs. ZLA, *p* < 0.05.

**Figure 5 dentistry-14-00030-f005:**
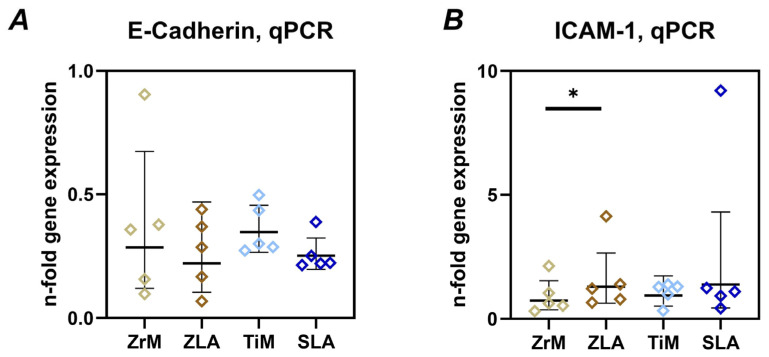
Gene expression of E-cadherin and intercellular adhesion molecule 1 (ICAM-1) in oral epithelial cells cultured on different surfaces. Ca9-22 cells were cultured on smooth machined zirconia (ZrM), sand-blasted acid-etched moderately rough zirconia (ZLA), smooth machined titanium (TiM), and sand-blasted acid-etched moderately rough titanium (SLA) surfaces for 7 days. The gene expression of E-cadherin (**A**) and ICAM-1 (**B**) was determined by qPCR. The data are presented as n-fold expression relative to tissue culture plastic (n = 1), calculated by the 2^−ΔΔCt^ method and using GAPDH as the housekeeping gene. Data are presented as mean and standard deviation (SD); each point represents a value obtained in an individual experiment. *—significant difference between ZrM vs. ZLA, *p* < 0.05.

## Data Availability

The original contributions presented in this study are included in the article. Further inquiries can be directed to the corresponding author.

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
