# Peer review of "Impact of Zirconia and Titanium Implant Surfaces of Different Roughness on Oral Epithelial Cells"

_dentistry, 2026, doi:10.3390/dj14010030_

Round 1
Reviewer 1 Report
Comments and Suggestions for Authors
The manuscript presents an in vitro study of human Ca9-22 epithelial cell line response to different zirconia and titanium implant surfaces. The scientific question is sound and important. Several methods were used to characterize surface characteristics in a previous article. The applied methods are up to date, just instead of static contact angle dynamic contact angle measurement method is more relevant in this case. The applied methods for determining cell proliferation are also good (CCK-8, fluorescent microscopy and gene expression). The study presents important and complex new data. Revisions must be done to enhance the quality of the manuscript. After revisions it will be appropriate for publication in Dentistry Journal.
- Materials and Methods:
- Statistical analysis: every data is an average value as measurement were performed on several discs and areas; therefore, SD is not enough to determine. Standard error of the mean (SE) will give the statistically relevant information when you compare average data. Please correct data and bar graphs, accordingly.
- Results:
- Please correct Figures 1, 3, 4 and 5 by replacing SD with SE.
- Discussion is well written and important references are discussed, but please insert the following article: Klinge B. and Meyle J.: Soft-tissue integration of implants. Consensus report of Working Group 2. Clin Oral Impl Res, Vol.17, pp.93–96, 2006
Reviewer 2 Report
Comments and Suggestions for Authors
A well designed methodology and results that need to be further investigated as well traslational studies. Hope to see these as soon as possible
Reviewer 3 Report
Comments and Suggestions for Authors
Dear authors,
Congratulations on your excellent work with your study !
While you refer readers to your previous article (line 114), I recommend including a summary of the surface characteristics of the specimens here !
Reviewer 4 Report
Comments and Suggestions for Authors
Respected authors,
It is my pleasure to evaluate an article such as this. Namely, this is a very interesting and useful article with the main aim of understanding the biological response to two types of dental materials. Research of this kind is highly recommended, as it helps fill gaps in the literature, and further studies in this area are important.
I also have a few suggestions regarding the organization of your research:
Abstract:
Please structure the Abstract using the following sections: Background, Aims, Materials, etc.
Include the p-value in the Results section of the Abstract.
Introduction:
Define all acronyms in this section, and then use them consistently throughout the rest of the manuscript.
Results:
Define all acronyms in the first figure, and use them consistently in subsequent figures.
Reviewer 5 Report
Comments and Suggestions for Authors
The manuscript titled " Impact of zirconia and titanium implant surfaces of different roughness on oral epithelial cells " addresses a relevant topic in implant dentistry: how implant surface roughness and material influence epithelial cell responses. The study provides orderly experimental work and clear data presentation; however, several aspects of the manuscript require improvement to meet academic publishing standards.
- The abstract gives a coherent overview of the study by outlining objectives, surface types, experimental methods, and major findings. However, it remains vague in quantitative terms, does not acknowledge key limitations, especially the use of Ca9-22 carcinoma cells, and at times overstates the functional relevance of modest in vitro differences.
- The keywords are accurate and align with the general scope of the study. Nevertheless, they are overly broad and could be more informative. The absence of specific mechanistic terms such as surface roughness, integrins, inflammation, or epithelial attachment reduces the precision and discoverability of the article.
- The introduction effectively underscores the clinical importance of peri-implant soft tissue sealing and cites pertinent literature. Yet, it does not fully reconcile conflicting evidence on epithelial responses to surface roughness, provides limited justification for selecting a carcinoma cell line, and lacks deeper mechanistic framing.
- The review is comprehensive and well referenced, demonstrating awareness of the broader scientific landscape. However, it summarizes prior findings more than it critically synthesizes them, missing opportunities to clarify why published results vary. Important concepts, e.g., surface energy, mechanotransduction, or advanced 3D tissue models, are underrepresented, which weakens the conceptual foundation for the study’s hypothesis.
- The methods are clearly described and generally reproducible, with adequate detail regarding surface preparation, assays, and statistical tests. Nonetheless, significant limitations exist. The use of Ca9-22 carcinoma cells severely limits physiological relevance; protein-level validation for adhesion markers is missing; surface characterization relies solely on prior publications rather than batch-specific measurements; and the experimental model does not incorporate microbial, inflammatory, or multicellular interactions essential to peri-implant soft tissue biology. These factors restrict the interpretability of the findings.
- The results are well structured with clear figures, and the discussion appropriately references existing literature and acknowledges some study constraints. However, many statistically significant findings have small biological effect sizes, and morphological observations are qualitative rather than quantitatively analyzed. Portions of the discussion interpret mRNA increases as functional enhancements without supporting protein-level or adhesion-strength data. Additionally, differences in roughness between ZLA and SLA surfaces confound the interpretation of material versus roughness effects, but this issue is underdeveloped in the discussion.
- The conclusions accurately restate the central observation that roughness exerts greater influence than material type. They also appropriately call for in vivo and primary-cell studies. However, the extrapolation to “stronger epithelial attachment” and improved sealing is premature without functional adhesion assays or microbial testing. The conclusions should more explicitly acknowledge the limitations arising from the simplified in vitro model and the roughness disparity between surface types.
The manuscript is recommended for publishing provided the above comments are adequately addressed.
Round 2
Reviewer 5 Report
Comments and Suggestions for Authors
The revision meets the reviewer's requirements, so the revised manuscript can be accepted for publication.
Comments on the Quality of English LanguageThe revision meets the reviewer's requirements, so the revised manuscript can be accepted for publication.